# Elevation of Pulmonary Artery Pressure in Newborns from High-Altitude Pregnancies Complicated by Preeclampsia

**DOI:** 10.3390/antiox12020347

**Published:** 2023-02-01

**Authors:** Carlos E. Salinas-Salmon, Carla Murillo-Jauregui, Marcelino Gonzales-Isidro, Vannia Espinoza-Pinto, Silvia V. Mendoza, Rosario Ruiz, Ronald Vargas, Yuri Perez, Jaime Montaño, Lilian Toledo, Abraham Badner, Jesús Jimenez, Javier Peñaranda, Catherine Romero, Martha Aguilar, Loyola Riveros-Gonzales, Ivar Arana, Eduardo Villamor

**Affiliations:** 1Instituto Boliviano de Biología de Altura (IBBA), UMSA, La Paz, Bolivia; 2Centro de Salud Tembladerani, La Paz, Bolivia; 3Hospital Materno Infantil, La Paz, Bolivia; 4Hospital de la Mujer, La Paz, Bolivia; 5Facultad de Medicina, UMSA, La Paz, Bolivia; 6Grupo Premio Nobel, La Paz, Bolivia; 7Maastricht University Medical Center (MUMC+), School for Oncology and Reproduction (GROW), 6202AZ Maastricht, The Netherlands

**Keywords:** high-altitude pregnancy, preeclampsia, pulmonary circulation, small for gestational age, fetal programming

## Abstract

We hypothesized that fetal exposure to the oxidative stress induced by the combined challenge of preeclampsia (PE) and high altitude would induce a significant impairment in the development of pulmonary circulation. We conducted a prospective study in La Paz (Bolivia, mean altitude 3625 m) in which newborns from singleton pregnancies with and without PE were compared (PE group *n* = 69, control *n* = 70). We conducted an echocardiographic study in these infants at the median age of two days. The percentage of cesarean deliveries and small for gestational age (SGA) infants was significantly higher in the PE group. Heart rate, respiratory rate, and oxygen saturation did not vary significantly between groups. Estimated pulmonary arterial pressure and pulmonary vascular resistance were 30% higher in newborns exposed to PE and high altitude compared with those exposed only to high altitude. We also detected signs of right ventricular hypertrophy in infants subjected to both exposures. In conclusion, this study provides evidence that the combination of PE and pregnancy at high altitude induces subclinical alterations in the pulmonary circulation of the newborn. Follow-up of this cohort may provide us with valuable information on the potential increased susceptibility to developing pulmonary hypertension or other pulmonary and cardiovascular disorders.

## 1. Introduction

Preeclampsia remains a leading cause of maternal and perinatal morbidity and mortality, affecting 2% to 8% of pregnancies worldwide [1]. Traditionally, preeclampsia is defined by the presence of elevated blood pressure and proteinuria, which identifies a group of pregnant women at high risk for adverse outcomes [2]. The etiopathogenesis of preeclampsia is complex and involves the interaction of numerous genetic, immunologic, and environmental factors [1,2]. Among the latter, pregnancy at high altitude (>2500 m) has been consistently linked with the development of preeclampsia [3,4,5,6,7]. There is a growing body of evidence that chronic hypoxemia due to high-altitude residence, operating on multiple maternal physiological characteristics, shifts the individual risk for the development of preeclampsia [3,4].

Although the increased incidence of preeclampsia is a significant contributor to high-altitude-induced intrauterine growth restriction (IUGR), IUGR may occur in high-altitude gestation in the absence of preeclampsia [4,7,8]. Therefore, the reduced oxygen availability that characterizes high altitude environments presents unavoidable, physiological challenges to pregnancy as well as fetal development [9]. In addition, either in association or separately, preeclampsia and IUGR have a profound impact on offspring health, both at birth and beyond the neonatal period. Preeclampsia, gestational hypoxia, and IUGR have been associated with an increased risk of cardiovascular diseases later in life, which is known as “fetal programming of adult diseases” or “developmental origins of health and disease” [4,10,11,12].

Pulmonary vascular dysfunction and hypertension have been reported as a common adverse neonatal outcome of both preeclampsia and high-altitude-induced IUGR [3,4,6,9,11,13,14,15,16,17,18,19,20,21]. Moreover, increasing evidence suggests a strong association between preeclampsia and/or IUGR and pulmonary hypertension in very preterm infants [22]. Placental oxidative stress and an altered balance of placenta-derived circulating pro- and anti-angiogenic factors are key factors in the pathophysiology of the pulmonary vascular alterations induced by preeclampsia and high-altitude pregnancy [4,11,15,16,18,23].

Therefore, pregnancy at high altitude represents an interesting in vivo paradigm to understand the importance of oxygen availability on fetal development, as well as to elucidate the pathogenesis of the effects of pregnancy complications in the offspring [8,19]. In the present study, we hypothesized that fetal exposure to the combined challenge of preeclampsia and high altitude would induce a significant impairment in the development of pulmonary circulation. To test this hypothesis, we conducted a prospective study in La Paz (Bolivia) in which newborns from 69 singleton pregnancies with preeclampsia were compared with newborns from 70 singleton pregnancies without hypertensive disorders.

## 2. Patients and Methods

This study was performed in the offspring of a group of pregnant women included in a study on the effects of preeclampsia on maternal pulmonary circulation. Data from the study in the mothers have been reported separately [24]. The sample size was calculated for the primary outcome (pulmonary hypertension in pregnant women with and without preeclampsia) [24] and not for the present secondary outcome. The study was carried out according to standards set by the Declaration of Helsinki and all procedures were approved by the Local Ethics Committee of the Bolivian Institute for High Altitude Biology (protocol code 07/2016, date of approval 17 August 2016, Consejo Técnico, Instituto Boliviano de Biología de Altura, Universidad Mayor de San Andrés, La Paz, Bolivia).

Women aged between 18 and 45 years old with a healthy pregnancy (control group) or a pregnancy complicated by preeclampsia were included in the study following their written consent. Preeclampsia was diagnosed according to guidelines set by the American College of Obstetrics and Gynecology [1]. All women participating in the study were highland residents for at least two generations, they currently resided in the city of La Paz (mean altitude 3625 m), did not move to lower altitudes during gestation, were non-smokers, had a normal body mass index (BMI), and no history of hypertension or other health complications. Multiple pregnancies were excluded. The pregnant women were recruited from five hospitals, all located in the city of La Paz (Bolivia): Hospital Materno-Infantil (Caja Nacional de la Salud), Hospital de la Mujer, Hospital Nuestra Señora de La Paz, Hospital Cotahuma, and Centro de Salud Materno Infantil Tembladerani. Although no specific data on socioeconomic level were collected, the population using these healthcare centers belonged to a lower-middle or lower economic level. The Gross Domestic Product (GDP) per capita of the department of La Paz in 2021 was USD 3777 (Bolivian GDP per capita: USD 3437) [25]. The human development index (HDI) of both the department of La Paz (0.719 in the year 2019) and Bolivia (0.718 in 2019) are considered high, as they are above 0.7 but below 0.8 [26].

At birth, data on the type of delivery, Apgar score, anthropometry, need for resuscitation, need for admission to the neonatal intensive care unit (NICU), and respiratory therapy were recorded. Small for gestational age (SGA) was defined as birth weight below the 10th percentile for sex and GA. On the day of the echocardiographic study, vital parameters (heart rate and respiratory rate), post-ductal oxygen saturation (Spo_2_), and end-tidal CO_2_ (EtCO_2_, measured by sidestream capnometry through nasal cannula) [27,28] were measured. An mCare 300 vital signs monitor (Spacelabs Healthcare, Snoqualmie, WA, USA) was used for the assessment of the vital parameters, Spo_2_, and capnometry.

### 2.1. Echocardiography Studies

All echocardiographic studies were performed by two cardiologists (CE S-S and PA) with extensive experience in neonatal cardiology. During the design phase of the study, these two investigators reached a consensus on the technical criteria for the echocardiographic measurements. The cardiologists were not involved in the clinical care of the infants and were blinded as to which study group the infants belonged to. Echocardiography studies were conducted in a warm environment in the quiet, awake state and with Doppler measurements across the pulmonary valve being made early in the study to avoid an elevation of estimated pressures because of agitation. No sedation was used, and infants were swaddled by parents or nursing staff.

All echocardiographic measurements were performed following the recommendations of the American Society of Echocardiography for the assessment of left ventricular (LV) and right ventricular (RV) geometry and function [29]. The echocardiography study included documentation of anatomy, presence of patent ductus arteriosus (PDA) and/or patent foramen ovale (PFO), morphometric measurements, and Doppler flow velocimetry values. The following parameters were measured: LV and RV end-diastolic dimension (derived in the parasternal long axis view); LV and RV end-systolic dimension (derived in the parasternal long axis view); LV and RV ejection fraction; RV midcavity diameter (derived in the four-chamber view); RV length (derived in the four-chamber view); RV wall thickness, right atrial (RA) end-systolic dimension; RA end-systolic area; RV myocardial performance index (Tei index); tricuspid annular plane systolic excursion (TAPSE); tricuspid annular systolic velocity (TASV index); tricuspid regurgitation (TR) peak systolic velocity (TR Vmax); and TR peak gradient.

Pulmonary artery pressure (PAP) was estimated by application of the modified Bernoulli equation: 4 × [TR Vmax]^2^ + expected RA pressure [30]. Pulmonary vascular resistance (PVR) was calculated as 10 × (TR Vmax/VTI_RVOT_) + 0.16 (VTI_RVOT_ = right ventricular outflow time-velocity integral) [31]. To mitigate the impact of body weight and size, we adjusted echocardiography morphometric values for body surface area (BSA) [29,32]. Linear parameters were related to the square root of BSA and area measurements were normalized directly to BSA [25,28]. BSA was calculated using the Haycock formula: BSA [m^2^] = 0.024265 × weight [kg]^0.5378^ × height [cm]^0.3964^ [33].

### 2.2. Statistics

Statistical analyses were conducted using JASP software version 0.16.3 [34]. Results for continuous variables are expressed as mean and standard deviation (SD) or, if variables were not normally distributed, as median and interquartile range (IQR). Data normality was tested using the Shapiro–Wilk test. Group comparisons were assessed by one-way ANOVA followed by Bonferroni’s post hoc *t*-test or Kruskal–Wallis with Bonferroni adjustment, as appropriate. Categorical variables are expressed as count and percentage and compared by the chi-square test. Differences were considered significant at a *p* < 0.05.

## 3. Results

Initially, 110 pregnant women with preeclampsia (PE group) and 98 pregnant women without preeclampsia (control group) were included in the study. In the control group, there was one fetal death and fifteen pregnant women dropped out of the study (eight women withdrew consent and seven did not attend the programmed controls). Seven additional pregnant women were excluded from the control group because they developed hypertension in late pregnancy, five because they showed signs of pulmonary pathology in the spirometry controls conducted during the study, and one because the pregnancy was twin. In the PE group, there were 12 fetal deaths and 28 pregnant women dropped out of the study (12 women withdrew consent and 16 did not attend the programmed controls). After birth, one additional infant was excluded due to the logistical impossibility of performing echocardiography in the first days of life. Therefore, the final cohort included 70 infants in the control group and 69 in the PE group. Maternal data have been reported separately [24]. Values for maternal age, height, weight, BMI, BSA, gravidity, and parity were similar for the control and PE women [24].

Of the 69 infants in the PE group, 35 were born before 37 weeks of GA. For the analysis of the results, the PE group was divided into two subgroups (PE-term and PE-preterm). Only one infant in the control group was born before 37 weeks GA and has been included in the analysis together with the control infants born at term. The baseline characteristics of the infants in the three groups are shown in Table 1. The percentage of cesarean deliveries and SGA were significantly higher in the PE groups than in the control group. After birth, five infants required resuscitation with intermittent positive pressure and/or supplemental oxygen. All these infants belonged to the PE-preterm group. Four of these infants required NICU admission and oxygen therapy by continuous positive airway pressure (CPAP) and one required oxygen supplementation in the incubator. On the day of the echocardiographic examination, none of the infants were receiving supplemental oxygen.

Considering only term infants, GA and birth weight were significantly lower in the PE group than in the control group (Table 1). The length at birth of the PE-term group was also lower than that of the control group but the difference was not statistically significant (*p* = 0.920). As expected, the PE-preterm group showed significantly lower gestational age, birth weight, and length than the two groups of full-term infants (Table 1).

The echocardiographic study was performed at a median age of 2 days (IQR 1–4 days) in the control group, 2 days (IQR 1–5 days) in the PE-term group, and 2 days (IQR 1–6 days) in the PE-preterm group. On the day of echocardiographic evaluation, average Spo_2_ values were within normal ranges for high-altitude-born infants and similar between groups (Table 1). There were also no statistically significant differences between the groups in heart or respiratory rate. As shown in Table 1, there was a statistically significant difference (*p* = 0.023) in the comparison of EtCO_2_ values between the control group and the PE-term group.

The echocardiographic data are shown in Table 2. The percentage of PDA was significantly higher in the both the PE-term and the PE-preterm groups when compared with the control group. The percentage of PFO was not significantly different among the three groups. Compared to the control group, infants in both the PE-term and PE-preterm groups showed a statistically significant increase in RA diameter, RV wall thickness, TR Vmax, TR peak gradient, PAP, and RVP. The TAPSE was significantly lower in the PE-preterm group when compared with both the PE-term and the control groups. The other echocardiographic measurements were not significantly different among the three groups.

## 4. Discussion

Evidence from cardiac catheterization, echocardiography, and electrocardiographic studies shows that the physiological postnatal decline in pulmonary artery pressure occurs more slowly in infants whose gestation and birth took place at high altitude [35,36,37]. Moreover, these infants appear to be at risk for persistence of fetal vascular connections, hypoxemia during infancy, and pulmonary hypertension [35,36]. However, there is a marked variability due to genetic adaptation, pregnancy complications, nutritional status, exposure to pollutants and toxins, socioeconomic status, and access to medical care [35,36]. The results of the present study suggest that fetal exposure to preeclampsia adds a relevant stress to the developing pulmonary vasculature in high-altitude gestation. Thus, during the first days of postnatal life, the estimated pulmonary arterial pressure and pulmonary vascular resistance were 30% higher in newborns exposed to preeclampsia and high altitude compared with a control group only exposed to high altitude. We also detected signs of right ventricular hypertrophy in infants subjected to both exposures. Our results confirm the higher pulmonary artery pressure values in preeclampsia-exposed infants reported by Heath-Freudenthal et al. in a very recent study also conducted in La Paz, Bolivia [11].

An important limitation of our study is that the echocardiographic evaluation of the infants was conducted at around 2 days of age. The reason for this was primarily logistical. Infants, particularly in the control group, did not stay long in the hospital, so echocardiography had to be performed in the first days of life. In newborns of mothers with preeclampsia, ultrasound was conducted in a time window comparable to that of the control group. Nevertheless, it cannot be ruled out that what we have observed in preeclampsia-exposed infants is merely a delayed circulatory transition without long-term effects. Moreover, the higher cesarean section rate in the preeclampsia group may have contributed to this delayed transition. In the study of Heath-Freudenthal et al., they conducted echocardiography at a later age (1 week of life) [11]. They observed that the pulmonary vascular effects of preeclampsia in the newborn were still present at that age, but were no longer present at the age of 6–9 months [11]. Therefore, the alterations in pulmonary circulation induced by the combination of preeclampsia and high-altitude gestation appears to decrease during the first months of life. Despite these results, Heath-Freudenthal et al. speculate that these children might be more susceptible to developing pulmonary hypertension later in life [11]. Interestingly, Jayet et al. reported that echocardiography-estimated pulmonary artery pressure in 14-year-old children born in La Paz and exposed to preeclampsia during gestation was 30% higher than those of age-matched children not exposed to preeclampsia [13].

Another important difference between our study and Heath-Freudenthal’s study is the clinical condition of the newborns. In our study, only five infants required resuscitation at birth and respiratory support during NICU admission. All these infants belonged to the PE-preterm group. In contrast, in the study of Heath-Freudenthal et al., a considerable percentage of both the preeclampsia and control groups required resuscitation and intensive care [11]. The main justification for this difference may be that our study included milder forms of preeclampsia. This difference is particularly interesting because it suggests that the alterations of neonatal pulmonary circulation are also present in the less severe forms of preeclampsia. Nevertheless, it should be noted that the echocardiographic findings in infants exposed to preeclampsia were not accompanied by relevant clinical alterations. Thus, vital parameters, including oxygen saturation, were not significantly different between infants exposed to high altitude plus preeclampsia and those only exposed to high altitude. The only exception was the EtCO_2_ values, which were slightly higher in the PE group than in the control group. EtCO_2_ measurement by sidestream capnometry through nasal cannula can provide an accurate and noninvasive estimate of PaCO_2_ levels in infants who are not intubated [27,28]. Moreover, the capnometry values obtained during the study were in agreement with those previously reported in newborns in La Paz [38]. The slight difference in EtCO_2_ values may reflect a higher respiratory distress in the PE group. However, infants in the PE group did not present greater tachypnea or more clinical signs of respiratory distress than infants in the control group. Therefore, the combination of preeclampsia and high-altitude gestation induced a subclinical increase in pulmonary arterial pressure.

On the other hand, the high number of fetal deaths that occurred in the PE group is a strong indicator of the pathological effect of the combination of preeclampsia and high altitude on the outcome of pregnancy [5]. Besides fetal mortality, there was a dropout rate of about 15% in the control group and 25% in the PE group. Although differential dropout is a problem that affects interventional more than observational studies [39], we cannot rule out the possibility that the health of mother–infant dyads that have dropped out of the study is different from that of those that have completed it. This could have introduced some unmeasurable biases in the results of our study.

A further limitation of our study is that in the absence of a control group at sea level we cannot draw any conclusions about the independent effect of high-altitude pregnancy on neonatal pulmonary circulation. However, this was not the aim of our study. Our objective was to analyze the effect of two potential challenges (preeclampsia and high altitude) vs. one (high altitude). It can be argued that the effects of preeclampsia would have been the same at sea level and, therefore, pregnancy at high altitude does not add any additional challenge. With our results we cannot refute that argument. However, one should consider the large body of literature (see [35,36] for review) that provides evidence for the pathogenic effect of high-altitude pregnancy when compared to pregnancy at sea level.

As mentioned above, a growing body of evidence supports that fetal stressors, such as high altitude and preeclampsia, acting at cellular and molecular levels, may alter pulmonary vascular development, leading to an increased vulnerability to pulmonary hypertension [20,21]. Although the pathophysiological mechanisms are not yet fully understood, substantial experimental and clinical data suggest that oxidative stress plays an important role in this process. In fact, a common feature of both preeclampsia and high-altitude gestation is increased oxidative stress [3,19,40,41]. In the first trimester, the placenta develops under a low-oxygen environment, which is essential for the conceptus who has little defense against reactive oxygen species (ROS) produced during oxidative metabolism [19,40]. Preeclampsia involves an abnormal placentation, characterized by a disturbed trophoblastic invasion of spiral arteries, which impairs the physiological increase in uteroplacental perfusion with advancing gestation. This may enhance the production of ROS, triggering a vicious cycle of oxidative stress, endothelial dysfunction, reduced placental perfusion, and hypoxia [19,40,41]. Interestingly, high-altitude placentas are similar to preeclamptic placentas in having a diminished antioxidant capacity and altered proliferation patterns of the uteroplacental vasculature [3,41,42]. Hypoxia-induced oxidative stress has been found to provide a strong stimulus for endoplasmic reticulum stress, leading to protein synthesis inhibition and impaired trophoblast proliferation and survival [35,37]. This has been considered a key mechanism in the pathogenesis of IUGR and pregnancy complications at high altitude [41,42].

Our study was limited to assessing clinical and cardiopulmonary function and therefore we can only speculate as to which pathophysiologic pathways are responsible for the elevation of pulmonary arterial pressure in newborns exposed to the double challenge of preeclampsia and high-altitude pregnancy. It has been suggested that placental hypoxia and/or oxidative stress produce alterations in the developing pulmonary vasculature through two main mechanisms: impaired angiogenesis and endothelial dysfunction. The rapid growth of the placenta is achieved by vasculogenic and angiogenic processes tightly regulated by a balance of pro-angiogenic factors, such as vascular endothelial growth factor (VEGF) and placental growth factor (PlGF), and anti-angiogenic factors, such as soluble fms-like tyrosine kinase-1(sFlt-1) and endoglin (sEng) [40,43,44,45,46]. In preeclamptic pregnancies, a dysregulation of the relative ratios of pro- and anti-angiogenic factors impairs the development of the placental vascular network [34] and may also affect the development of the fetal pulmonary vasculature [44]. Accordingly, preclinical studies showed that intra-amniotic exposure to sFlt-1 induced sustained pulmonary vascular and alveolar abnormalities in the offspring [44,47]. Our study did not include the assessment of pro- or anti-angiogenic factors, but Heath-Freudenthal et al. reported in their study that maternal and umbilical cord blood sFlt1 levels were higher in preeclampsia and positively associated with elevated pulmonary artery pressure [11].

Regarding endothelial dysfunction, the two most extensively studied endothelial mediators are nitric oxide (NO) and endothelin-1, ET-1. An imbalance between these two essential endothelial agonists is implicated in a large number of cardiovascular pathological conditions, including the impairment of uteroplacental perfusion in preeclamptic and high-altitude pregnancies as well as in pulmonary hypertension [7,41,48,49,50,51,52,53]. NO is a key vasodilator and antiproliferative mediator in both the uteroplacental and pulmonary circulation, and a decreased production and/or increased scavenging of NO disrupts vascular development and function. ET-1 interacts with NO by altering gene expression and ligand–receptor interactions, providing a close link between NO and ET-1 signaling, and generating a powerful vasoconstrictor and mitogenic effect [41,50,53,54,55,56]. Both hypoxia and oxidative stress are key in the pathogenesis of these endothelial alterations [41,56].

## 5. Conclusions

In summary, the present study, together with that of Heath-Freudenthal et al. [11], provides evidence that the combination of preeclampsia and pregnancy at high altitude induces subclinical alterations in the pulmonary circulation of the newborn. Follow-up of our cohort, as well as of similar ones, over the next few years may provide us with valuable information on the potential increased susceptibility of developing pulmonary hypertension or other pulmonary and cardiovascular disorders.

## Figures and Tables

**Table 1 antioxidants-12-00347-t001:** Characteristics and vital parameters of the included infants.

	Group A: Control *n* = 70	Group B: PE-Term *n* = 37	*p*-Value A vs. B	Group C: PE-Preterm *n* = 32	*p*-Value A vs. C	*p*-Value B vs. C
Gestational age (weeks)	39.6 [38.7–40.3]	38.4 [37–40]	0.007	34.9 [32.4–35.4]	<0.001	<0.001
Birth weight (kg)	3.24 (0.41)	2.90 (0.45)	0.003	2.19 (0.68)	<0.001	<0.001
SGA	7 (10.0%)	11 (29.7%)	0.009	9 (28.1%)	0.020	0.884
Length (cm)	48.2 (5.8)	47.7 (3.6)	0.920	44.4 (4.7)	0.002	0.018
Male sex	40 (57.1%)	20 (54.1%)	0.759	17 (53.1%)	0.705	0.938
Cesarean section	27 (38.6%)	31 (83.8%)	<0.001	28 (87.5%)	<0.001	0.662
Apgar 1	8 [7–8]	8 [7–8]	0.283	7 [7–8]	0.013	0.180
Apgar 5	9 [9–9]	9 [9–9]	0.238	9 [9–9]	0.019	0.433
Resuscitation	0	0	-	5 (15.6%)	<0.001	<0.001
NICU admission	0	0	-	5 (15.6%)	<0.001	<0.001
Heart rate	128.8 (17.2)	126.9 (18.3)	0.868	138.1 (19.3)	0.045	0.031
Respiratory rate	51.4 (12.4)	47.4 (13.1)	0.266	52.4 (13.1)	0.927	0.236
Spo_2_ (%)	90.1 (4.4)	88.8 (3.8)	0.253	91.7 (3.8)	0.176	0.052
EtCO_2_ (mmHg)	25.7 (5.4)	28.8 (6.1)	0.023	27.0 (6.2)	0.821	0.381

Results are expressed as count (%), mean (SD), or median [25th–75th percentile]. EtCO_2_: end-tidal carbon dioxide; NICU: neonatal intensive care unit; SGA: small for gestational age.

**Table 2 antioxidants-12-00347-t002:** Echocardiographic parameters.

	Group A: Control *n* = 70	Group B: PE-Term *n* = 37	*p*-Value A vs. B	Group C: PE-Preterm *n* = 32	*p*-Value A vs. C	*p*-Value B vs. C
Patent ductus arteriosus	11 (15.7%)	14 (37.8%)	0.010	12 (37.5%)	0.015	0.977
Patent foramen ovale	3 (4.3%)	1 (2.7%)	0.681	3 (9.4%)	0.311	0.237
LV end-diastolic dimension ^a^ (cm)/$\sqrt{\mathrm{BSA}}$	4.43 (0.55)	4.17 (0.44)	0.040	4.04 (0.59)	0.002	0.570
LV end-systolic dimension ^a^ (cm)/$\sqrt{\mathrm{BSA}}$	2.82 (0.44)	2.66 (0.41)	0.169	2.61 (0.42)	0.063	0.880
LV ejection fraction (%)	67.9 (5.0)	67.2 (5.1)	0.808	67.5 (5.1)	0.932	0.976
$\mathrm{RA}\;\mathrm{length}\;(\mathrm{cm})/\sqrt{\mathrm{BSA}}$	2.91 (2.25)	2.77 (0.27)	0.029	2.76 (0.28)	0.028	0.993
RA end-systolic area (cm^2^)/BSA	8.51 (1.55)	8.23 (1.10)	0.675	8.23 (2.03)	0.607	0.267
RV end-diastolic dimension ^a^ (cm)/$\sqrt{\mathrm{BSA}}$	1.84 (0.32)	1.79 (0.26)	0.625	1.77 (0.30)	0.508	0.977
RV end-systolic dimension ^a^ (cm)/$\sqrt{\mathrm{BSA}}$	1.37 (1.15)	1.14 (0.20)	0.379	1.23 (0.22)	0.710	0.906
RV ejection fraction (%)	67.7 (4.7)	67.3 (4.2)	0.882	67.5 (4.7)	0.974	0.976
RV midcavity diameter ^b^ (cm)/$\sqrt{\mathrm{BSA}}\;$	2.21 (0.25)	2.25 (0.30)	0.776	2.31 (0.26)	0.197	0.602
$\mathrm{RV}\;\mathrm{length}{\;}^{b}\;(\mathrm{cm})/\sqrt{\mathrm{BSA}}$	3.75 (0.40)	3.73 (0.32)	0.960	3.80 (0.66)	0.895	0.805
$\mathrm{RV}\;\mathrm{wall}\;\mathrm{thickness}\;(\mathrm{cm})/\sqrt{\mathrm{BSA}}$	0.46 (0.09)	0.56 (0.12)	<0.001	0.54 (0.11)	0.002	0.817
RV Tei index	0.35 (0.03)	0.34 (0.04)	0.062	0.35 (0.03)	0.777	0.394
TAPSE (cm)	0.84 (0.11)	0.83 (0.14)	0.981	0.75 (0.12)	0.005	0.023
TASV (m.s^−1^)	0.07 [0.06–0.08]	0.07 [0.06–0.08]	0.122	0.07 [0.05–0.08]	0.053	0.482
TR Vmax (m.s^−1^)	2.5 [2.3–2.6]	3.0 [2.8–3.2]	<0.001	2.9 [2.8–3.2]	<0.001	0.499
TR peak gradient (mmHg)	25.0 [22.3–27.8]	36.0 [32.0–42.0]	<0.001	33.0 [31.0–40.3]	<0.001	0.440
PAP (mmHg)	35.07 (3.83)	46.68 (5.44)	<0.001	45.69 (6.22)	<0.001	0.683
PVR (Wood units)	2.23 [2.00–2.71]	2.99 [2.58–3.53]	<0.001	2.84 [2.59–3.29]	<0.001	0.576

Results are expressed as count (%), mean (SD), or median [25th–75th percentile]. BSA: body surface area (m^2^); LV: left ventricle; PAP: pulmonary artery pressure; PVR: pulmonary vascular resistance; RA: right atria; RV: right ventricle; TAPSE: tricuspid annular plane systolic excursion; TASV: tricuspid annular systolic velocity; Tei index: myocardial performance index, TR: tricuspid regurgitation; Vmax: peak systolic velocity. ^a^ Derived in the parasternal long axis view. ^b^ Derived in the four-chamber view.

## Data Availability

All data relevant to the study are included in the article. Additional data are available upon reasonable request.

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
