# Peer review of "Elevation of Pulmonary Artery Pressure in Newborns from High-Altitude Pregnancies Complicated by Preeclampsia"

_antioxidants, 2023, doi:10.3390/antiox12020347_

Round 1
Reviewer 1 Report
The MS describes an interesting concept, it has the potential to contribute to the pathomechanism of preeclampsia during pregnancies, as living at high altitude with low oxygen supply might be a relevant risk factor for pregnant women and their babies.
A weak point is the claim in the last sentence of the abstract. "These alterations may render infants more ... xxx ... . (xxx = whatever you want). If there are any data to link prenatal hypoxia with pulmonary hypertension in adulthood, it should be included in the MS. Without any data on "prenatal programming" the risk for pulmonary hypertension, the statement abitrarily picks just one possibility out of hundreds.
To increase the strength of the MS I suggest several additions and text modifications, some of them necessary to make the message comprehensible to a range of readers.
In methods I miss information on the altitude the mothers live. Do all Patients in Bolivia live at the same altitude? Is there a group difference, or is it compareable?
In line 86 the authors mentions : "... women should be highland residents for at least two generations ... ",
- why should be, and not "were ..." please provide some information on the altitude your study participants (both groups) come from.
What was the socio-economic status and some characteristics like Body-mass-index of the mothers between the groups? If socio-economic status or life style is a risk-factor for PE, this may constitute a relevant bias to be kept in mind when discussing the results.
It is irritating to have the mothers data in reference [24]. Ideally it should be summarized here be comprehensible for the reader. Searching for [24] I only found a not peer-reviewed-version. If you insist not to provide these data in this MS, It should be indicated in the bibliography of this MS whether or not the reference [24] is published or at least accepted.
Results
Why "infant death" is an explanation for drop out, and not part of the group comparison, please discuss.
How large was the drop out rate for specific reasons, in particular "incomplete assessment" in and between the groups ? was the drop out big enough to constitute a reason for bias ? please report and discuss.
Table 1:
a legend to explain abbreviations and technical terms would be helpful. (similar as done in table 2)
there are two colums "P vs. Control" please clarify
what is are the compared groups under "P vs. control" (twice) and what are the compared groups under "P vs. PE-Term", please clarify (mention both groups the P is associated with) in table 1 and table 2.
In table 2 the legend (I missed it in Table 1) is very helpful but inconsistent, it needs to be revisited.
- most parameters have no Unit, please complete
- some parameters (e.g. RVL) are explained but difficult to find, If RV means right ventricle, why not move the parameters starting with RV together in one block and explain it once at the start of the block?
- RVP presuably should be PVR, please correct the typo
Discussion
line 197: what is a "special risk" ? more than a "risk"?
I miss the discussion on the possible impact of the relatively high drop out rate, what is the chance for a systematic drop out related bias/difference between control and PE-patients?
line 197 to 198: "infants appear to be at special risk for persistence .... pulmonary hypertension." Where does this information come from? please cite/referene unequivocally.
Line 205-206: "We also detected signs of right ventricular ... exposed to both insults." What do you mean by both insults? Is living at high altitude an "Insult"?
Line 206: "Our results confirm those reported by ....". Do you mean all results? or some results? Please specify which ones; e.g. "Our results on x, y, z (Table 2), confirm those reported by ... "
line 259-261"Since the placenta and the developing lung have many structural and functional similarities, both organs may have similar responses to alterations in the intrauterine environment during pregnancy [38]."
This statement is not only inplausible, I can not see any relevant similarity between placenta and the developing lung. Furthermore, this statement is NOT corroborated by the citation [38]. Why is it here?
line 297, please remove the "." before [9]
line 314: I did not see any "supplementary information" if there is any, it should be mentioned/refered to in the text.
Author Response
Reviewer 1
- The MS describes an interesting concept, it has the potential to contribute to the pathomechanism of preeclampsia during pregnancies, as living at high altitude with low oxygen supply might be a relevant risk factor for pregnant women and their babies.
A weak point is the claim in the last sentence of the abstract. "These alterations may render infants more ... xxx ... . (xxx = whatever you want). If there are any data to link prenatal hypoxia with pulmonary hypertension in adulthood, it should be included in the MS. Without any data on "prenatal programming" the risk for pulmonary hypertension, the statement arbitrarily picks just one possibility out of hundreds.
Response: Thank you very much for your thoughtful review of the manuscript and constructive criticisms. Following your suggestion, we have replaced the last sentence of the abstract with the following:
“Follow-up of this cohort may provide us with valuable information on the potential increased susceptibility to develop pulmonary hypertension or other pulmonary and cardiovascular disorders.”
- To increase the strength of the MS I suggest several additions and text modifications, some of them necessary to make the message comprehensible to a range of readers.
In methods, I miss information on the altitude the mothers live. Do all Patients in Bolivia live at the same altitude? Is there a group difference, or is it comparable?
Response: The information in the methods chapter has been rewritten to include information on the altitude of the city of La Paz and the socioeconomic status of the pregnant women included in the study. The following paragraph has been included:
“All women participating in the study were highland residents for at least two generations, they currently resided in the city of La Paz (mean altitude 3625 m), did not move to lower altitudes during gestation, were non-smokers, had a normal body mass index (BMI), and no history of hypertension or other health complications. Multiple pregnancies were ex-cluded. The pregnant women were recruited from five hospitals, all located in the city of La Paz (Bolivia): Hospital Materno-Infantil (Caja Nacional de la Salud), Hospital de la Mujer, Hospital Nuestra Señora de La Paz, Hospital Cotahuma, and Centro de Salud Materno Infantil Tembladerani. Although no specific data on socio-economic level were collected, the population using these healthcare centers belonged to a lower-middle or lower economic level. The Gross Domestic Product (GDP) per capita of the department of La Paz in 2021 was US$ 3777 (Bolivian GDP per capita: US$ 3437) [25]. The human de-velopment index (HDI) of both the department of La Paz (0.719 in the year 2019) and Bo-livia (0.718 in 2019) are considered high as they are above 0.7 but below 0.8 [26]. “
- In line 86 the authors mentions : "... women should be highland residents for at least two generations ... ",
- why should be, and not "were ..." please provide some information on the altitude your study participants (both groups) come from.
This issue has been addressed above.
- What was the socio-economic status and some characteristics like Body-mass-index of the mothers between the groups? If socio-economic status or life style is a risk-factor for PE, this may constitute a relevant bias to be kept in mind when discussing the results.
See above for the socioeconomic status. BMI was not significantly different between the PE-group and the control group. This information is now provided in the new version of the manuscript.
- It is irritating to have the mothers data in reference [24]. Ideally it should be summarized here be comprehensible for the reader. Searching for [24] I only found a not peer-reviewed-version. If you insist not to provide these data in this MS, It should be indicated in the bibliography of this MS whether or not the reference [24] is published or at least accepted.
We apologize for this problem which has now been solved as the mothers' data have been published in a preprint that is cited in the bibliography replacing the previous reference 24 that was only an abstract. In addition, the new version of the manuscript specifically mentions the absence of significant differences in age or anthropometry between the control group and the preeclampsia group.
- Results
Why "infant death" is an explanation for drop out, and not part of the group comparison, please discuss.
How large was the drop out rate for specific reasons, in particular "incomplete assessment" in and between the groups ? was the drop out big enough to constitute a reason for bias ? please report and discuss.
Response: The correct expression would have been fetal death instead of infant death. The following paragraph has been included in the Results chapter of the new version of the manuscript:
“Initially, 110 pregnant women with preeclampsia (PE-group) and 98 pregnant women without preeclampsia (control group) were included in the study. In the control group, there was one fetal death and 15 pregnant women dropped out of the study (eight women withdrew consent and seven did not attend the programmed controls). Seven ad-ditional pregnant women were excluded from the control group because they developed hypertension in late pregnancy, five because they showed signs of pulmonary pathology in the spirometry controls conducted during the study, and one because pregnancy was twin. In the PE-group, there were 12 fetal deaths and 28 pregnant women dropped out of the study (12 women withdrew consent and 16 did not attend the programmed controls). After birth, one additional infant was excluded due to the logistical impossibility of per-forming echocardiography in the first days of life. Therefore, the final cohort included 70 infants in the control group and 69 in the PE-group. Maternal data have been reported separately [24]. Values for maternal age, height, weight, BMI, BSA, gravidity, and parity were similar for control and PE women [24].”
- Table 1:
a legend to explain abbreviations and technical terms would be helpful. (similar as done in table 2)
there are two colums "P vs. Control" please clarify
what is are the compared groups under "P vs. control" (twice) and what are the compared groups under "P vs. PE-Term", please clarify (mention both groups the P is associated with) in table 1 and table 2.
In table 2 the legend (I missed it in Table 1) is very helpful but inconsistent, it needs to be revisited.
- most parameters have no Unit, please complete
- some parameters (e.g. RVL) are explained but difficult to find, If RV means right ventricle, why not move the parameters starting with RV together in one block and explain it once at the start of the block?
- RVP presumably should be PVR, please correct the typo
Response: Units have been included in the new version of the manuscript.
The tables have been modified and additional the groups are now identified as group A: Control, group B: Preeclampsia-term, and group C: Preeclampsia-preterm. In this way it is clearer what the different comparisons are (A vs. B, A vs. C, and B vs. C).
The number of abbreviations has also been reduced to make the table easier to read without recurring so often to the explanatory legend. Finally, following your suggestion, we have grouped the right ventricle and left ventricle data together.
The abbreviation PVR has been corrected.
- Discussion
line 197: what is a "special risk" ? more than a "risk"?
The word “special” has been removed.
- I miss the discussion on the possible impact of the relatively high drop out rate, what is the chance for a systematic drop out related bias/difference between control and PE-patients?
The following paragraph has been included in the manuscript:
“On the other hand, the high number of fetal deaths that occurred in the PE-group is a strong indicator of the pathological effect of the combination of preeclampsia and high al-titude on the outcome of pregnancy [5]. Besides fetal mortality, there was a dropout rate of about 15% in the control group and 25% in the PE-group. Although differential dropout is a problem that affects interventional more than observational studies [39], we cannot rule out the possibility that the health of mother-infant dyads that have dropped out of the study is different from that of those that have completed it. This could have introduced some unmeasurable bias in the results of our study.”
- Line 197 to 198: "infants appear to be at special risk for persistence .... pulmonary hypertension." Where does this information come from? please cite/referene unequivocally.
This information was taken from the reviews by Niermeyer et al (references 35 and 36 in the new version of the manuscript).
- Line 205-206: "We also detected signs of right ventricular ... exposed to both insults." What do you mean by both insults? Is living at high altitude an "Insult"?
The term insult has been removed in the new version of the manuscript.
- Line 206: "Our results confirm those reported by ....". Do you mean all results? or some results? Please specify which ones; e.g. "Our results on x, y, z (Table 2), confirm those reported by ... "
We have been more specific and have rewritten the paragraph as follows:
“Our results confirm the higher pulmonary artery pressure values in preeclampsia–exposed infants reported by Heath-Freudenthal et al. in a very recent study also conducted in La Paz, Bolivia [11].”
- line 259-261"Since the placenta and the developing lung have many structural and functional similarities, both organs may have similar responses to alterations in the intrauterine environment during pregnancy [38]."
This statement is not only inplausible, I can not see any relevant similarity between placenta and the developing lung. Furthermore, this statement is NOT corroborated by the citation [38]. Why is it here?
The statement about the similarities between the placenta and the developing lung has been withdrawn in the new version of the manuscript.
- Line 297, please remove the "." before [9]
Removed.
- Line 314: I did not see any "supplementary information" if there is any, it should be mentioned/refered to in the text.
There is no supplementary information and therefore the reference to it has been eliminated in the new version of the manuscript.
Reviewer 2 Report
It is unclear what the role of "high altitude" is since no data at the sea level are presented.
Since this manuscript has been submitted to the journal Antioxidants, data on oxidative stress parameters and/or antioxidant levels, for example, in the plasma may be desirable.
Author Response
Reviewer 2
- It is unclear what the role of "high altitude" is since no data at the sea level are presented.
Response: Thank you very much for your comments. Our study aimed to evaluate the potential additive pathologic effect of preeclampsia on pregnancy at high altitude. A sea level study group would have been necessary if the aim had been to evaluate separately the effect of high altitude and preeclampsia on neonatal pulmonary circulation. There are several studies that have investigated this and are cited in the references of our manuscript. Our design was based on a control group subjected to a single stress (high altitude) and a second group exposed to a double stress (high altitude and preeclampsia). In fact, the only precedent in the literature of a study similar to ours (Heath-Freudenthal et al, reference 11 in the manuscript) also did not include pregnant women at sea level.
- Since this manuscript has been submitted to the journal Antioxidants, data on oxidative stress parameters and/or antioxidant levels, for example, in the plasma may be desirable.
Response: We agree that this is an important limitation of our manuscript and we acknowledge this in the article. We would have liked to complete our study with an evaluation of some markers of oxidative stress but unfortunately, this is beyond the economic/technological reach of our research group. However, there is overwhelming evidence of the central pathophysiological role of oxidative stress in both high-altitude gestation and preeclampsia. This led us to submit the manuscript to "Antioxidants". We believe that our manuscript may be of interest to the readers of "Antioxidants" and hope that the editors of the journal will consider our article to be within the scope of the journal.
Reviewer 3 Report
I have read this paper with great interest, and value the work as provided. However, and with a background on both neonatology and DOHAD, I do have concerns that the colleagues are overinterpreting their data, as they also somewhat indicate in the discussion section. Data have been collected on day 2, so during transition, not necessary similar to long lasting effects, but perhaps rather reflecting delayed adaptation ? related to this, ‘development of the pulmonary circulation’, and the conclusions in the abstract are overstretching what the authors have observed.
‘insult’ is also perhaps a too strong wording, especially for high altitude. Perhaps a more neutral wording like factor or covariate is more appropriate ?
Was there an a priori power calculation done, as there was already some literature on high altitude ? Otherwise, the study is rather explorative.
Suggest to add the technical information on the etCO2 equipment (assuming that this was not mCare vital signs ?). Along the same line, and reading the CO2 results in the table 1, is this a sufficiently useful and reliable technique in neonates ?
How were investigators truly blinded, as eg SGA incidence is quite different and some were preterm ? were images read post hoc or measurement checked (interrater variability)
Table 2 suggest that the majority of findings were distributed normally ?
Were differences between the preterm and term PE cases explored ? and was type of delivery explored ? (cesarean is only mentioned in the abstract, but the data are not discussed in the paper, unless I have missed this)
Editing reflection: a conclusions usually does not contain citations ? and the conclusions in the full version reads much more balanced compared to the abstract’s conclusion.
As the requested revisions are 'moderate', i have selected to go for major to be able to reassess the revised version.
Author Response
Reviewer 3
- I have read this paper with great interest, and value the work as provided. However, and with a background on both neonatology and DOHAD, I do have concerns that the colleagues are overinterpreting their data, as they also somewhat indicate in the discussion section. Data have been collected on day 2, so during transition, not necessary similar to long lasting effects, but perhaps rather reflecting delayed adaptation ? related to this, ‘development of the pulmonary circulation’, and the conclusions in the abstract are overstretching what the authors have observed.
Response: Thank you very much for your thoughtful review of the manuscript and constructive criticisms. In the new version of the manuscript, we clearly acknowledge that our findings may simply reflect a delayed circulatory transition in preeclampsia-exposed infants:
“Nevertheless, it cannot be ruled out that what we have observed in preeclampsia-exposed infants is merely a delayed circulatory transition without long-term effects.”
In addition, the conclusion of the abstract has been tempered as follows:
“In conclusion, this study provides evidence that the combination of PE and pregnancy at high altitude induces subclinical alterations in the pulmonary circulation of the newborn. Follow-up of this cohort may provide us with valuable information on the potential increased susceptibility to develop pulmonary hypertension or other pulmonary and cardiovascular disorders”
- ‘insult’ is also perhaps a too strong wording, especially for high altitude. Perhaps a more neutral wording like factor or covariate is more appropriate?
Response: Following your suggestion, we have removed the term "insult" from the manuscript.
- Was there an a priori power calculation done, as there was already some literature on high altitude ? Otherwise, the study is rather explorative.
Response: The primary study was on development of pulmonary hypertension in pregnant women (reference 24). It is for this primary outcome that the sample size was calculated. This issue is now clearly specified in the new version of the manuscript.
- Suggest to add the technical information on the etCO2 equipment (assuming that this was not mCare vital signs ?). Along the same line, and reading the CO2 results in the table 1, is this a sufficiently useful and reliable technique in neonates ?
Response: The mCare 300 monitor has the option to perform capnometry measurements. Regarding the reliability of the technique, the following sentences have been included in the new version of the manuscript:
“Thus, vital parameters, including oxygen saturation, were not significantly different between infants exposed to high altitude plus preeclampsia and those only exposed to high altitude. The only exception was the EtCO2 values, which were slightly higher in the PE group than in the control group. EtCO2 measurement by sidestream capnometry through nasal cannula can provide an accurate and noninvasive estimate of PaCO2 levels in infants who are not intubated [27, 28]. Moreover, the capnometry values obtained during the study were in agreement with those previously reported in newborns in La Paz [38]. The slight difference in EtCO2 values may reflect a higher respiratory distress in the PE group. However, infants in the PE group did not present greater tachypnea or more clinical signs of respiratory distress than infants in the control group. Therefore, the combination of preeclampsia and high altitude gestation induced a subclinical increase in pulmonary arterial pressure.”
- How were investigators truly blinded, as eg SGA incidence is quite different and some were preterm ? were images read post hoc or measurement checked (interrater variability).
We agree that absolute blinding was impossible, particularly with regard to preterm infants who all belonged to the preeclampsia group. In term newborns it was easier because the weight differences were not so marked. The following information has been included in the manuscript:
“All echocardiographic studies were performed by two cardiologists (CE S-S and PA) with extensive experience in neonatal cardiology. During the design phase of the study, these two investigators reached a consensus on the technical criteria for the echocardiographic measurements. The cardiologists were not involved in the clinical care of the infants and were blinded as to which study group the infants belonged to.”
- Table 2 suggest that the majority of findings were distributed normally ?
As mentioned in the methods, data normality was tested using the Shapiro-Wilk test. Indeed, most of the data followed a normal distribution.
- Were differences between the preterm and term PE cases explored ? and was type of delivery explored ?
Differences between preterm and term PE were also explored (B vs. C in the new version of tables).
- Cesarean rate is also mentioned (cesarean is only mentioned in the abstract, but the data are not discussed in the paper, unless I have missed this)
Response: Cesarean section rate was higher in the preeclampsia group. This is mentioned in the abstract and in the results (text and Table 1). In the new version of the manuscript the higher rate of cesarean section is briefly discussed:
“An important limitation of our study is that the echocardiographic evaluation of the infants was conducted at around 2 days of age. The reason for this was primarily logistical. Infants, particularly in the control group, did not stay long in the hospital, so echocardiography had to be performed in the first days of life. In newborns of mothers with preeclampsia, ultrasound was conducted in a time window comparable to that of the control group. Nevertheless, it cannot be ruled out that what we have observed in preeclampsia-exposed infants is merely a delayed circulatory transition without long-term effects. Moreover, the higher cesarean section rate in the preeclampsia group may have contributed to this delayed transition.”
9.Editing reflection: a conclusions usually does not contain citations ? and the conclusions in the full version reads much more balanced compared to the abstract’s conclusion.
Response: We agree that references are generally not cited in the conclusions, but in this case it is necessary for a better understanding of the message. The citation of Heath-Freudenthal et al is necessary since our results and theirs have been discussed together throughout the manuscript and the conclusion is based on the combination of the results of both studies. The other citation has been removed. In addition, and as mentioned above, we have tempered the conclusion in the abstract.
Reviewer 4 Report
Summary:
The authors prospectively conducted an echocardiographic study in newborns from singleton pregnancies with and without preeclampsia (PE) at the median age of two days. They found that estimated pulmonary arterial pressure and pulmonary vascular resistance were 30% higher in newborns exposed to PE and high altitudes. The authors conclude that a combination of PE and pregnancy at high altitude induces subclinical alterations in the pulmonary circulation of the newborn.
General concerns:
1. Line103: Please delete “in” in “Echocardiography studies were conducted in in a warm environment in the quiet” and check the words throughout the manuscript.
2. Abstract: Lines 28-29: Heart rate, respiratory rate, oxygen saturation and 28 end-tidal CO2 did not vary significantly between groups. Lines 167-168: “There were also no statistically significant differences between the groups in heart rate, respiratory rate, or end-tidal CO2”. However, the PE-Term group exhibited significantly higher Et CO2 than the control group in Table 1. Please discuss higher Et CO2 this finding.
3. Line 189: Please delete ‘()’ and replace with ‘;’ in “TAPSE: tricuspid annular plane systolic excursion ()”.
4. Discussion: Lines 219-221: “However, it has been suggested that these pulmonary vascular alterations may render infants more vulnerable to develop pulmonary hypertension during childhood and/or adult life [11]”. The article describe “Preeclampsia augmented fetal hypoxia and increased the risk of PH in the neonate but not later in infancy” in Abstract. Please clarify this.
5. Abstract: Conclusions: “These alterations may render infants more vulnerable to develop pulmonary hypertension during childhood and/or adult life”. The authors did not measure pulmonary hypertension in these subjects during childhood and/or adult life. Please provide more evidence to support the conclusions.
Author Response
Reviewer 4
General concerns:
- Line103: Please delete “in” in “Echocardiography studies were conducted in in a warm environment in the quiet” and check the words throughout the manuscript.
Thank you very much for your thoughtful review of the manuscript and constructive criticisms. This typo has been corrected
- Abstract: Lines 28-29: Heart rate, respiratory rate, oxygen saturation and 28 end-tidal CO2 did not vary significantly between groups. Lines 167-168: “There were also no statistically significant differences between the groups in heart rate, respiratory rate, or end-tidal CO2”. However, the PE-Term group exhibited significantly higher Et CO2 than the control group in Table 1. Please discuss higher Et CO2 this finding.
This was indeed an error in the first version of the manuscript that has now been corrected. The differences in EtCO2 have been discussed as follows:
“Thus, vital parameters, including oxygen saturation, were not significantly different between infants exposed to high altitude plus preeclampsia and those only exposed to high altitude. The only exception was the EtCO2 values, which were slightly higher in the PE group than in the control group. EtCO2 measurement by sidestream capnometry through nasal cannula can provide an accurate and noninvasive estimate of PaCO2 levels in infants who are not intubated [27, 28]. Moreover, the capnometry values obtained during the study were in agreement with those previously reported in newborns in La Paz [38]. The slight difference in EtCO2 values may reflect a higher respiratory distress in the PE group. However, infants in the PE group did not present greater tachypnea or more clinical signs of respiratory distress than infants in the control group. Therefore, the combination of preeclampsia and high altitude gestation induced a subclinical increase in pulmonary arterial pressure.”
- Line 189: Please delete ‘()’ and replace with ‘;’ in “TAPSE: tricuspid annular plane systolic excursion ()”.
Corrected.
- Discussion: Lines 219-221: “However, it has been suggested that these pulmonary vascular alterations may render infants more vulnerable to develop pulmonary hypertension during childhood and/or adult life [11]”. The article describe “Preeclampsia augmented fetal hypoxia and increased the risk of PH in the neonate but not later in infancy” in Abstract. Please clarify this.
The paragraph has been rewritten as follows:
An important limitation of our study is that the echocardiographic evaluation of the infants was conducted at around 2 days of age. The reason for this was primarily logisti-cal. Infants, particularly in the control group, did not stay long in the hospital, so echocar-diography had to be performed in the first days of life. In newborns of mothers with preeclampsia, ultrasound was conducted in a time window comparable to that of the control group. Nevertheless, it cannot be ruled out that what we have observed in preeclampsia-exposed infants is merely a delayed circulatory transition without long-term effects. Moreover, the higher cesarean section rate in the preeclampsia group may have contributed to this delayed transition. In the study of Heath-Freudenthal et al., they conducted echocardiography at a later age (1 week of life) [11]. They observed that the pulmonary vascular effects of preeclampsia in the newborn were still present at that age but were no longer so at the age of 6-9 months [11]. Therefore, the alterations in pulmonary circulation induced by the combination of preeclampsia and high-altitude gestation appears to decrease during the first months of life. Despite these results, Heath-Freudenthal et al. speculate that these children might be more susceptible to developing pulmonary hypertension later in life [11]. Interestingly, Jayet et al. reported that echocardiography-estimated pulmonary artery pressure in 14-year-old children born in La Paz and exposed to preeclampsia during gestation was 30% higher than those of age-matched children not exposed to preeclampsia [13].
- Abstract: Conclusions: “These alterations may render infants more vulnerable to develop pulmonary hypertension during childhood and/or adult life”. The authors did not measure pulmonary hypertension in these subjects during childhood and/or adult life. Please provide more evidence to support the conclusions.
This speculative part of the conclusion has been eliminated in the new version of the manuscript.
Round 2
Reviewer 2 Report
Please discuss in the manuscript concerning "It is unclear what the role of high altitude is since no data at the sea level are presented." Since this study does not indicate the effects of the high altitude, the statement like "Estimated pulmonary arterial pressure and pulmonary vascular resistance were 30% higher in newborns exposed to PE and high altitude compared with those exposed only to the second insult high altitude." is not appropriate.
Author Response
- Please discuss in the manuscript concerning "It is unclear what the role of high altitude is since no data at the sea level are presented." Since this study does not indicate the effects of the high altitude, the statement like "Estimated pulmonary arterial pressure and pulmonary vascular resistance were 30% higher in newborns exposed to PE and high altitude compared with those exposed only to the second insult high altitude." is not appropriate.
Response: Thank you very much for your comments. Following your suggestion, we have discussed this limitation as follows:
“A further limitation of our study is that in the absence of a control group at sea level we cannot draw any conclusions about the independent effect of high altitude pregnancy on neonatal pulmonary circulation. However, this was not the aim of our study. Our objective was to analyze the effect of two potential challenges (preeclampsia and high altitude) vs. one (high altitude). It can be argued that the effects of preeclampsia would have been the same at sea level and, therefore, pregnancy at high altitude does not add any additional challenge. With our results we cannot refute that argument. However, one should consider the large body of literature (see [35, 36] for review) that provides evidence for the pathogenic effect of high altitude pregnancy when compared to pregnancy at sea level.”
We have also rewritten the sentence you mention as follows:
“Thus, during the first days of postnatal life, the estimated pulmonary arterial pressure and pulmonary vascular resistance were 30% higher in newborns exposed to preeclampsia and high altitude compared with a control group only exposed to high altitude.”
We honestly believe that with our experimental design and results we can keep that sentence. It is made very clear that the control situation was pregnancy at high altitude. We are just saying that if the control population (high altitude) is exposed to preeclampsia there is a 30% increase in pulmonary artery pressure. We honestly believe that our results support that statement.
Reviewer 3 Report
no additional comments, suggest to accept
Author Response
Thank you very much.
Reviewer 4 Report
The authors have addressed the concerns.
Round 3
Reviewer 2 Report
.